# LEARNING TEMPORAL EVOLUTION OF PROBABILITY DISTRIBUTION WITH RECURRENT NEURAL NETWORK

## ABSTRACT

We propose to tackle a time series regression problem by computing temporal evolution of a probability density function to provide a probabilistic forecast. A Recurrent Neural Network (RNN) based model is employed to learn a nonlinear operator for temporal evolution of a probability density function. We use a softmax layer for a numerical discretization of a smooth probability density functions, which transforms a function approximation problem to a classification task. Explicit and implicit regularization strategies are introduced to impose a smoothness condition on the estimated probability distribution. A Monte Carlo procedure to compute the temporal evolution of the distribution for a multiple-step forecast is presented. The evaluation of the proposed algorithm on three synthetic and two real data sets shows advantage over the compared baselines.

## 1 INTRODUCTION

Application of the deep learning for manufacturing processes has attracted a great attention as one of the core technologies in Industry 4.0 (Lasi et al., 2014). In many manufacturing processes, e.g. blast furnace, smelter, and milling, the complexity of the overall system makes it almost impossible or impractical to develop a simulation model from the first principles. Hence, system identification from sensor observations has been a long-standing research topic (Wang et al., 2016). Still, when the observation is noisy and there is no prior knowledge on the underlying dynamics, there is only a very limited number of methods for the reconstruction of nonlinear dynamics.

In this work, we consider the following class of problems, where the system is driven by a complex underlying dynamical system, e.g.,

$$\frac{\partial y}{\partial t} = \mathcal{F}(y(t), y(t - \tau), \boldsymbol{u}(t)). \tag{1}$$

Here, $y(t)$ is a continuous process, $\mathcal{F}$ is a nonlinear operator, $\tau$ is a delay-time parameter, and $\boldsymbol{u}(t)$ is an exogenous forcing, such as control parameters. At time step $t$, we then observe a noisy measurement of $y(t)$ which can be defined by the following noise model

$$\hat{y}_t = y(t)\nu_t + \epsilon_t, \tag{2}$$

where $\nu_t$ is a multiplicative and $\epsilon_t$ is an additive noise process. In (1) and (2), we place no assumption on function $\mathcal{F}$, do not assume any distributional properties of noises $\nu_t$ and $\epsilon_t$, but assume the knowledge of the control parameters $\boldsymbol{u}(t)$.

Since the noise components, $\nu_t$ and $\epsilon_t$, are stochastic processes, the observation $\hat{y}_t$ is a random variable. In this work, we are interested in computing temporal evolution of the probability density function (PDF) of $\hat{y}$, given the observations up to time step $t$, i.e., $p(\hat{y}_{t+n}|\widehat{\boldsymbol{Y}}_{0:t}, \boldsymbol{U}_{0:t+n-1})$ for $n \geq 1$, where $\widehat{\boldsymbol{Y}}_{0:t} = (\hat{y}_0, \cdots, \hat{y}_t)$ is a trajectory of the past observations and $\boldsymbol{U}_{0:t+n-1} = (\boldsymbol{u}_0, \cdots, \boldsymbol{u}_{t+n-1})$ consists of the history of the known control actions, $\boldsymbol{U}_{0:t-1}$, and a future control scenario, $\boldsymbol{U}_{t:t+n-1}$. We show, in Section 3, a class of problems, where simple regression problem of forecasting the value of $\hat{y}_{t+n}$ is not sufficient or not possible, e.g., chaotic systems. Note that the computation of time evolution of a PDF has been a long-standing topic in statistical physics. For a simple Markov process, there are well-established theories based on the Fokker-Planck equation. However, it is very difficult to extend those theories to a more general problem, such as delay-time dynamical systems, or apply it to complex nonlinear systems.

Modeling of the system (1) has been extensively studied in the past, in particular, under the linearity assumptions on $\mathcal{F}$ and certain noise models, e.g., Gaussian $\epsilon_t$ and $\nu_t = 1$ in (2). The approaches based on auto-regressive processes (Lütkepohl, 2005) and Kalman filter (Harvey, 1990) are good examples. Although these methods do estimate the predictive probability distribution and enable the computation of the forecast uncertainty, the assumptions on the noise and linearity in many cases make it challenging to model real nonlinear dynamical systems.

Recently, a nonlinear state-space model based on the Gaussian process, called the Gaussian Process State Space Model (GPSSM), has been extended for the identification of nonlinear system (Frigola et al., 2013; Eleftheriadis et al., 2017). GPSSM is capable of representing a nonlinear system and is particularly advantageous when the size of the data set is relatively small that it is difficult to train a deep learning model. However, the joint Gaussian assumption of GPSSM may restrict the representation capability for a complex non-Gaussian noise.

A recent success of deep learning created a flurry of new approaches for time series modeling and prediction. The ability of deep neural networks, such as RNN, to learn complex nonlinear spatio-temporal relationships in the data enabled these methods to outperform the classical time series approaches. For example, in the recent works of Qin et al. (2017); Hsu (2017); Dasgupta & Osogami (2017), the authors proposed different variants of the RNN-based algorithms to perform time series predictions and showed their advantage over the traditional methods. Although encouraging, these approaches lack the ability to estimate the probability distribution of the predictions since RNN is a deterministic model and unable to fully capture the stochastic nature of the data.

To enable RNN to model the stochastic properties of the data, Chung et al. (2015) augmented RNN with a latent random variable included in the hidden state and proposed to estimate the resulting model using variational inference. In a similar vein, the works of Archer et al. (2015); Krishnan et al. (2017) extend the traditional Kalman filter to handle nonlinear dynamics when the inference becomes intractable. Their approach is based on formulating the variational lower bound and optimizing it under the assumption of Gaussian posterior.

Another recent line of works enabled stochasticity in the RNN-based models by drawing a connection between Bayesian variation inference and a dropout technique. In particular, Gal & Ghahramani (2016) showed that the model parameter uncertainty (which then leads to uncertainty in model predictions), that traditionally was estimated using variational inference, can be approximated using a dropout method (a random removal of some connections in the network structure). The prediction uncertainty is then estimated by evaluating the model outputs at different realizations of the dropout weights. Following the ideas of Gal & Ghahramani (2016), Zhu & Laptev (2017) proposed additional ways (besides modeling the parameter uncertainty) to quantify the forecast uncertainty in RNN, which included the model mis-specification error and the inherent noise of the data.

## 1.1 OVERVIEW OF THE PROPOSED WORK

We propose an RNN-model to compute the temporal evolution of a PDF, $p(\hat{y}_{t+n}|\widehat{\boldsymbol{Y}}_{0:t}, \boldsymbol{U}_{0:t+n-1})$. To avoid the difficulties in directly estimating the continuous function, we use a numerical discretization technique, which converts the function approximation problem to a classification task (see Section 2.2). We note that the use of the traditional cross-entropy (CE) loss in our formulated classification problem can be problematic since it is oblivious to the class ordering. To address this, we additionally propose two regularizations for CE to account for a geometric proximity between the classes (see Sections 2.2.1 and 2.2.2). The probability distribution of one-step-ahead prediction, $p(\hat{y}_{t+1}|\widehat{\boldsymbol{Y}}_{0:t}, \boldsymbol{U}_{0:t})$ can now be simply estimated from the output softmax layer of RNN (see Section 2.2), while to propagate the probability distribution further in time, for a multiple-step forecast, we propose a sequential Monte Carlo (SMC) method (see Section 2.4). For clarity, we present most derivations for univariate time series but also show the extension to multivariate data in Section 2.3. We empirically show that the proposed modeling approach enables us to represent a continuous PDF of any arbitrary shape, including the ability to handle the multiplicative data noises in (2). Since the probability distribution is computed, the RNN-model can also be used for a regression task by computing the expectation (see Section 2.4). Hereafter, we use DE-RNN for the proposed RNN model, considering the similarity with the density-estimation task.

In summary, the contributions of this work are as follows: (i) formulate the classical regression problem for time series prediction as a predictive density-estimation problem, which can be solved

by a classification task (ii) propose an approach to compute the time evolution of probability distribution using SMC on the distributions from DE-RNN (iii) proposed two regularizations for CE loss to capture the ordering of the classes in the discretized PDF. We evaluated the proposed algorithm on three synthetic and two real datasets, showing its advantage over the baselines. Note that DE-RNN has a direct relevance to a wide range of problems in physics and engineering, in particular, for uncertainty quantification and propagation (Zhang & Karniadakis, 2017).

## 2 LSTM FOR NOISY DYNAMICAL SYSTEM

In this Section we present the details of the proposed algorithm using a specific form of RNN, called Long Short-Term Memory (LSTM) network. Although in the following presentation and experiments we used LSTM, other networks, e.g., GRU (Chung et al., 2014), can be used instead.

### 2.1 REVIEW OF LONG SHORT-TERM MEMORY NETWORK

The Long Short-Term Memory network (LSTM) (Hochreiter & Schmidhuber, 1997; Gers et al., 2000) consists of a set of nonlinear transformations of input variables $\boldsymbol{z}_t \in \mathbb{R}^m$;

$$\text{Gating functions: } \boldsymbol{G}_{i,f,o} = \boldsymbol{\varphi}_S \circ \mathcal{L}(\boldsymbol{z}_t), \tag{3}$$

$$\text{Internal state: } \boldsymbol{s}_t = (\boldsymbol{1} - \boldsymbol{G}_f) \odot \boldsymbol{s}_{t-1} + \boldsymbol{G}_i \odot (\boldsymbol{\varphi}_T \circ \mathcal{L}(\boldsymbol{z}_t)), \tag{4}$$

$$\text{Output: } \boldsymbol{h}_t = \boldsymbol{G}_o \odot \boldsymbol{s}_t. \tag{5}$$

Here, $\boldsymbol{\varphi}_S$ and $\boldsymbol{\varphi}_T$, respectively, denote the sigmoid and hyperbolic tangent functions, $\mathcal{L}$ is a linear layer, which includes a bias, $\boldsymbol{s}_t \in \mathbb{R}^{N_c}$ is the internal state, $\boldsymbol{h}_t \in \mathbb{R}^{N_c}$ is the output of the LSTM network, $N_c$ is the number of the LSTM units, and $\boldsymbol{a} \odot \boldsymbol{b}$ denote a component-wise multiplication.

Interesting observation can be made about equation (4). We can re-write equation (4) as

$$\boldsymbol{s}_{t+1} = [1 - f(\boldsymbol{z}_t)] \, \boldsymbol{s}_t + g(\boldsymbol{z}_t), \tag{6}$$

for some functions $f$ and $g$. With a simple re-scaling, this equation can be interpreted as a first-order Euler scheme for a linear dynamical system,

$$\frac{d\boldsymbol{s}}{dt} = -f(\boldsymbol{z})\boldsymbol{s} + g(\boldsymbol{z}). \tag{7}$$

Thus, LSTM can be understood as a series expansion, where a complex nonlinear dynamical system is approximated by a combination of many simpler dynamical systems.

Usually, LSTM network is supplemented by feed-forward neural networks, e.g.,

$$\boldsymbol{z}_t = \mathcal{F}_{in}(\boldsymbol{x}_t, \boldsymbol{h}_{t-1}), \quad \boldsymbol{P}_{t+1} = \mathcal{F}_{out}(\boldsymbol{h}_t), \tag{8}$$

in which $\boldsymbol{x}_t$ is the input feature. Using (5), we can denote by $\Psi_e$ and $\Psi_d$ a collection of the operators from input to internal state (encoder) and from internal state to the output $\boldsymbol{P}$ (decoder):

$$\boldsymbol{s}_t = \Psi_e(\boldsymbol{x}_t, \boldsymbol{s}_{t-1}), \quad \boldsymbol{P}_{t+1} = \Psi_d(\boldsymbol{s}_t). \tag{9}$$

### 2.2 DISCRETE APPROXIMATION OF PROBABILITY DENSITY FUNCTION

In this Section we first consider the problem of modeling the conditional PDF, $p(\hat{y}_{t+1}|\widehat{\boldsymbol{Y}}_{0:t}, \boldsymbol{U}_{0:t})$. Although $\hat{y}_{t+1}$ has a dependence on the past trajectories of both $\hat{y}$ and $\boldsymbol{u}$, using the "state space" LSTM model argument in Section 2.1, the conditional PDF can be modeled as a Markov process

$$p(\hat{y}_{t+1}|\widehat{\boldsymbol{Y}}_{0:t}, \boldsymbol{U}_{0:t}) = p(\hat{y}_{t+1}|\hat{y}_t, \boldsymbol{u}_t, \boldsymbol{s}_{t-1}) = p(\hat{y}_{t+1}|\boldsymbol{s}_t). \tag{10}$$

Hence, to simplify the problem, we consider a task of estimating the PDF of a random variable $\hat{y}$, given an input $x$, i.e., $p(\hat{y}|x)$. The obtained results can then be directly applied to the original problem of estimating $p(\hat{y}_{t+1}|\boldsymbol{s}_t)$.

Let $\boldsymbol{\alpha} = (\alpha_0, \cdots, \alpha_K)$ denote a set of real numbers, such that $\alpha_{i-1} < \alpha_i$ for $i = 1, \cdots, K$, which defines $K$ disjoint intervals, $\mathcal{I}_i = (\alpha_{i-1}, \alpha_i)$. Then, a discrete probability distribution can be defined

$$p(k|x) = \int_{\mathcal{I}_k} p(\hat{y}|x)dy, \text{ for } k = 1, \dots, K, \tag{11}$$

where it is clear that $p(k|x)$ is a numerical discretization of the continuous PDF, $p(\hat{y}|x)$. Using the LSTM from Section 2.1, the discrete probability $p(k|x)$ can be modeled by the softmax layer ($P$) as an output of $\Psi_d$ in (9) such that

$$p(k|x) = P_k, \text{ for } k = 1, \dots, K. \tag{12}$$

Thus, the original problem of estimating a smooth function, $p(\hat{y}|x)$, is transformed into a classification problem of estimating $p(k|x)$ in a discrete space. Obviously, the size of the bin, $|\mathcal{I}_j|$, affects the fidelity of the approximation. The effects of the bin size are presented in Section 3.1. There is a similarity between the discretization and the idea of Lin et al. (2007). However, it should be noted that the same discretization technique, often called "finite volume method", has been widely used in the numerical simulations of partial differential equations for a long time.

The discretization naturally leads to the conventional cross-entropy (CE) minimization. Suppose we have a data set, $\boldsymbol{D}_R = \{(\hat{y}_i, x_i); \hat{y}_i \in \mathbb{R}, x_i \in \mathbb{R}, \text{ and } i = 1, \dots, N\}$. We can define a mapping $\mathcal{C} : \mathbb{R} \to \mathbb{N}_+$ such that $\mathcal{C}(\hat{y}) = k$, if $y \in \mathcal{I}_k$. Then, $\boldsymbol{D}_R$ can be easily converted to a new data set for target labels, $\boldsymbol{D}_C = \{(c_i, \hat{y}_i, x_i); c_i \in \mathbb{N}_+, \hat{y}_i \in \mathbb{R}, x_i \in \mathbb{R}, \text{ and } i = 1, \dots, N\}$, where $c_i = \mathcal{C}(\hat{y}_i)$. $\boldsymbol{D}_C$ provides a training data set for the following CE minimization problem,

$$CE = -\sum_{n=2}^{N} \sum_{k=1}^{K} \delta_{c_n k} \log P_k^n = -\sum_{n=2}^{N} \log P_{c_n}^n. \tag{13}$$

Note, however, that the CE minimization does not explicitly guarantee the smoothness of the estimated distribution. Since CE loss function depends only on $P_i$ of a correct label, $\delta_{c_n k}$, as a result, in the optimization problem every element $P_i$, except for the one corresponding to the correct label, $P_{c_n}$, is penalized in the same way, which is natural in the conventional classification tasks where a geometric proximity between the classes is not relevant. In the present study, however, the softmax layer, or class probability, is used as a discrete approximation to a smooth function. Hence, it is expected that $P_{c_n}$ and $P_{c_n \pm 1}$ (i.e., the nearby classes) should be close to each other. To address this issue, in the following Sections 2.2.1 and 2.2.2, we propose two types of regularization to impose the class proximity structure in the CE loss.

### 2.2.1 EXPLICIT REGULARIZATION OF CROSS-ENTROPY LOSS

To explicitly impose the smoothness between the classes, we propose to use a regularized cross-entropy (RCE) minimization, defined by the following loss function

$$\text{RCE} = \sum_{n=2}^{N} \left\{ \sum_{k=1}^{K} -\delta_{c_n k} \log P_k^n + \lambda \left(\boldsymbol{L}\boldsymbol{P}^n\right)^T \boldsymbol{L}\boldsymbol{P}^n \right\}, \tag{14}$$

where $\lambda$ is a penalty parameter and the Laplacian matrix $\boldsymbol{L} \in \mathbb{R}^{K-2, K}$ is

$$\boldsymbol{L} = \begin{bmatrix} 1 & -2 & 1 & 0 & \cdots & 0 \\ 0 & 1 & -2 & 1 & \cdots & 0 \\ \cdots\cdots\cdots\cdots\cdots\cdots\cdots\cdots\cdots \\ 0 & \cdots & 0 & 1 & -2 & 1 \end{bmatrix}. \tag{15}$$

RCE is analogous to the penalized maximum likelihood solution for density estimation (Silverman, 1986). Assuming a uniform bin size, $|\mathcal{I}_0| = \cdots = |\mathcal{I}_K| = \delta y$, the Laplacian of a distribution can be approximated by a Taylor expansion $p''(\hat{y}|x)|_{y=\alpha_{i-1/2}} \simeq (P_{i-1} - 2P_i + P_{i+1})/\delta y^2$, where $\alpha_{i-1/2} = 0.5(\alpha_{i-1} + \alpha_i)$. Then, it is clear that

$$\left(\boldsymbol{L}\boldsymbol{P}^n\right)^T \boldsymbol{L}\boldsymbol{P}^n \sim \int \left[p''(\hat{y}|x)\right]^2 dy. \tag{16}$$

In other words, RCE aims to smooth out the distribution by penalizing local minima or maxima.

### 2.2.2 IMPLICIT REGULARIZATION OF CROSS-ENTROPY LOSS

Alternative to adding an explicit regularization to CE, the smoothness can be achieved by enforcing a spatial correlation in the network output. Here, we use an one-dimensional convolution layer to

enforce smoothness. Let $\widetilde{\boldsymbol{o}} \in \mathbb{R}^K$ denote the last layer of DE-RNN, which was the input to the softmax layer. We can add a convolution layer, $\boldsymbol{o} \in \mathbb{R}^K$, on top of $\widetilde{\boldsymbol{o}}$, such that

$$o_i = \sum_{j=1}^{K} \frac{1}{h} \exp\left[-\frac{1}{2}\left(\frac{i-j}{h}\right)^2\right] \widetilde{o}_j, \ \ \text{for} \ i = 1, \cdots, K, \tag{17}$$

where the parameter $h$ determines the smoothness of the estimated distribution. Then, $\boldsymbol{o}$ is supplied to the softmax layer. Using (17), DE-RNN can now be trained by the standard CE. The implicit regularization, here we call convolution CE (CCE), is analogous to a kernel density estimation.

## 2.3 Multivariate Time Series

In the modeling of multivariate time series, it is usually assumed that the noise is independent, i.e., the covariance of the noise is a diagonal matrix. In this case, it is straightforward to extend DE-RNN, by using multiple softmax layers as the output of DE-RNN. However, such an independent noise assumption significantly limits the representative capability of an RNN. Here, we propose to use a set of independently trained DE-RNNs to compute the joint PDF of a multivariate time series.

Let $\hat{\boldsymbol{y}}_t$ be a $l$-dimensional multivariate time series; $\hat{\boldsymbol{y}}_t = (\hat{y}_t^{(1)}, \cdots, \hat{y}_t^{(l)})$. The joint PDF can be represented by a product rule,

$$p(\hat{\boldsymbol{y}}_{t+1}) = p\left(\hat{y}_{t+1}^{(l)}|\hat{y}_{t+1}^{(l-1)}, \cdots, \hat{y}_{t+1}^{(1)}\right) p\left(\hat{y}_{t+1}^{(l-1)}|\hat{y}_{t+1}^{(l-2)}, \cdots, \hat{y}_{t+1}^{(1)}\right) \cdots p\left(\hat{y}_{t+1}^{(2)}|\hat{y}_{t+1}^{(1)}\right) p\left(\hat{y}_{t+1}^{(1)}\right),$$

where the dependency on the past trajectory $(\widehat{\boldsymbol{Y}}_{0:t}, \boldsymbol{U}_{0:t})$ is omitted in the notation for simplicity. Directly learning the joint PDF, $p(\hat{\boldsymbol{y}}_{t+1}|\widehat{\boldsymbol{Y}}_{0:t}, \boldsymbol{U}_{0:t})$, in a tensor product space is not scalable. Instead, a set of DE-RNN is trained to represent the conditional PDFs shown on the right hand side of the above expression. Then, the joint PDF can be computed by a product of the Softmax outputs of the DE-RNNs. Note that, although it requires training $l$ DE-RNNs to compute the full joint PDF, there is no dependency between the DE-RNNs in the training phase. So, the set of DE-RNNs can be trained in parallel, which can greatly reduce the training time. The details of the multivariate DE-RNN are explained in Appendix A.

## 2.4 Computing Time Evolution of Probability Distribution

The inputs to a DE-RNN are $(\hat{y}_t, \boldsymbol{u}_t)$, and the output is the probability distribution,

$$\boldsymbol{P}_{t+1} = \Psi_d(\boldsymbol{s}_t) = \Psi_d \circ \Psi_e(\hat{y}_t, \boldsymbol{u}_t, \boldsymbol{s}_{t-1}).$$

Note that $\boldsymbol{D}_C$ is used only in the training stage. Then, the moments of the predictive distribution can be easily evaluated, e.g.,

$$E[\hat{y}_{t+1}|\widehat{\boldsymbol{Y}}_{0:t}, \boldsymbol{U}_{0:t}] = \boldsymbol{\alpha}_{1/2}^T \boldsymbol{P}_{t+1}, \ Var[\hat{y}_{t+1}|\widehat{\boldsymbol{Y}}_{0:t}, \boldsymbol{U}_{0:t}] = (\boldsymbol{\alpha}_{1/2}^2)^T \boldsymbol{P}_{t+1} - E[\hat{y}_{t+1}|\widehat{\boldsymbol{Y}}_{0:t}, \boldsymbol{U}_{0:t}]^2, \tag{18}$$

where $\boldsymbol{\alpha}_{1/2} = (\alpha_{1/2}, \alpha_{1+1/2}, \cdots, \alpha_{K-1/2})^T$, $\boldsymbol{\alpha}_{1/2}^2 = \boldsymbol{\alpha}_{1/2} \odot \boldsymbol{\alpha}_{1/2}$, and $\alpha_{i-1/2} = 0.5(\alpha_{i-1} + \alpha_i)$.

Next, we consider a multiple-step forecast, which corresponds to computing a temporal evolution of the probability distribution, i.e., $p(\hat{y}_{t+n}|\widehat{\boldsymbol{Y}}_{0:t}, \boldsymbol{U}_{0:t+n-1})$ for $n > 1$. For simplicity, the multiple-step forecast is shown only for a univariate time series. An extension to a multivariate time series is straightforward (Appendix A).

Applying the results of Section 2.2, once the distribution of $\hat{y}_{t+1}$ in (10) is computed, the distribution of $\hat{y}_{t+2}$ can be similarly obtained as $p(\hat{y}_{t+2}|\boldsymbol{s}_{t+1})$. Observe that $\boldsymbol{s}_{t+1}$ is computed from a deterministic function of $\boldsymbol{s}_t$, $\boldsymbol{u}_{t+1}$, and $\hat{y}_{t+1}$, i.e.,

$$\boldsymbol{s}_{t+1} = \Psi_e(\hat{y}_{t+1}, \boldsymbol{u}_{t+1}, \boldsymbol{s}_t).$$

Here, $\boldsymbol{u}_{t+1}$ and $\boldsymbol{s}_t$ are already known, while $\hat{y}_{t+1}$ is a random variable, whose distribution $p(\hat{y}_{t+1}|\boldsymbol{s}_t)$ is computed from the deterministic function $\Psi_d(\boldsymbol{s}_t)$. Then, $\boldsymbol{s}_{t+1}$ is also a random variable. The distribution, $p(\boldsymbol{s}_{t+1}|\boldsymbol{s}_t, \boldsymbol{u}_{t+1})$, can be obtained by applying a change of variables on $p(\hat{y}_{t+1}|\boldsymbol{s}_t)$ with

a nonlinear mapping $\Psi_e$. Repeating this process, the multiple-step-ahead predictive distribution can therefore be computed as

$$p(\hat{y}_{t+n}|\widehat{\boldsymbol{Y}}_{0:t}, \boldsymbol{U}_{0:t+n-1}) = \int \cdots \int p(\hat{y}_{t+n}|\boldsymbol{s}_{t+n-1}) \prod_{i=1}^{n-1} p(\boldsymbol{s}_{t+i}|\boldsymbol{s}_{t+i-1}, \boldsymbol{u}_{t+i})\, d\boldsymbol{s}_{t+i}. \qquad (19)$$

Since the high dimensional integration in (19) is intractable, we propose to approximate it by a sequential Monte Carlo method. The Monte Carlo procedure is outlined in Algorithm 1.

---

**Algorithm 1** Sequential Monte Carlo method for LSTM multi-step-ahead prediction

---

**Input**: $\widehat{\boldsymbol{Y}}_{0:t}$, $\boldsymbol{U}_{0:t}$, number of Monte Carlo samples, $N_s$, and forecast horizon $n$

**Output**: $p(\hat{y}_{t+n}|\widehat{\boldsymbol{Y}}_{0:t}, \boldsymbol{U}_{0:t+n-1})$ (density estimation from $\hat{\boldsymbol{y}}_{t+n}$)

Initialization: Set LSTM states to $\boldsymbol{s}_0 = \boldsymbol{h}_0 = \boldsymbol{0}$

Perform a sequential update of LSTM up to time $t$ from the noisy observations ($\widehat{\boldsymbol{Y}}_{0:t}$).

$$\boldsymbol{s}_i = \Psi_e(\hat{y}_i, \boldsymbol{u}_i, \boldsymbol{s}_{i-1}) \text{ for } i = 1, \cdots, t.$$

Make $N_s$ replicas of the internal state, $\boldsymbol{s}_t^1 = \cdots = \boldsymbol{s}_t^{N_s} = \boldsymbol{s}_t$.

**repeat**

Compute the predictive distribution of $\hat{y}_{t+1}^i$ for each sample

$$\boldsymbol{P}_{t+1}^i = \Psi_d(\boldsymbol{s}_t^i), \text{ for } i = 1, \cdots, N_s.$$

Sample the target variable at $t+1$, $\hat{y}_{t+1}^i$, from the distribution

    1.    Sample the class label from the discrete distribution: $c^i \sim \boldsymbol{P}_{t+1}^i$

    2.    Sample $\hat{y}_{t+1}^i$ in $\mathcal{I}_{c^i}$: $\hat{y}_{t+1}^i \sim \mathcal{U}(\alpha_{c^i-1}, \alpha_{c^i})$

Update the internal state of LSTM

$$\boldsymbol{s}_{t+1}^i = \Psi_e(\hat{y}_{t+1}^i, \boldsymbol{u}_{t+1}, \boldsymbol{s}_t^i).$$

**until** (all $\hat{\boldsymbol{y}}_{t+n}$ are sampled)

---

## 3 EXPERIMENTS

In this section, DE-RNN is tested against three synthetic and two real data sets. The LSTM architecture used in all of the numerical experiments is identical. Two feed-forward networks are used before and after the LSTM;

$$\boldsymbol{z}_t = \mathcal{L}\left(\varphi_T \circ \mathcal{L}(y_t, \boldsymbol{u}_t) + \boldsymbol{h}_{t-1}\right), \qquad \boldsymbol{P}_{t+1} = \varphi_{SM} \circ \mathcal{L}(\varphi_T \circ \mathcal{L}(\varphi_{SP} \circ \mathcal{L}(\boldsymbol{h}_t))), \qquad (20)$$

in which $\varphi_{SP}$ and $\varphi_{SM}$ denote the softplus and softmax functions, respectively. The size of the bins is kept uniform, i.e., $|\mathcal{I}_1| = \cdots = |\mathcal{I}_K| = \delta y$. The LSTM is trained by using ADAM (Kingma & Ba, 2015) with a minibath size of 20 and a learning rate of $\eta = 10^{-3}$.

### 3.1 COX-INGERSOLL-ROSS PROCESS

First, we consider a modified Cox-Ingersoll-Ross (CIR) process, which is represented by the following stochastic differential equation,

$$dy(t) = -0.5y(t)dt + \sqrt{0.5 + |y(t)|}dW, \qquad (21)$$

in which $W$ is the Weiner process. The original CIR process is used to model the valuation of interest rate derivatives in finance. Equation (21) is solved by the forward Euler method with the time step size $\delta t = 0.1$. The simulation is performed for $T = (0, 160000]\delta t$ to generate the training data and $T = (160000, 162000]\delta t$ is used for the testing. Note that the noise component of CIR is multiplicative, which depends on $y(t)$.

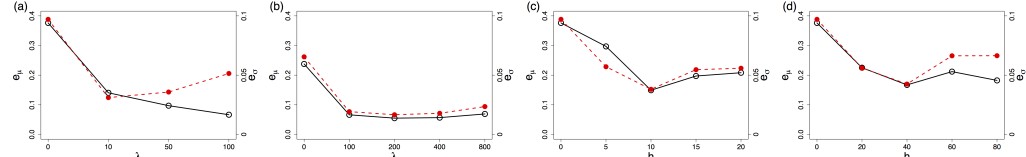

Figure 1: NRMSE of the next-step prediction of the CIR process by RCE (a,b) and CCE (c,d). The bin size is (a,c) $\delta y = 0.08$ and (b,d) 0.04. The hollow circles ($\circ$) denote NRMSE in the expectation ($e_\mu$) and the solid circles ($\bullet$) are the standard deviation ($e_\sigma$).

Table 1: NRMSE of the mean and standard deviation of the next-step prediction. The DE-RNN results are compared with the first-order autoregressive model (AR), Kalman filter (KF), and Gaussian process (GP).

|  | CE | RCE | CCE | AR(1) | KF | GP |
|---|---|---|---|---|---|---|
| $e_\mu$ | 0.238 | 0.0549 | 0.149 | 0.029 | 0.029 | 0.831 |
| $e_\sigma$ | 0.066 | 0.017 | 0.038 | 0.228 | 0.228 | 0.095 |

The experiments are performed for two different bin sizes, $dy = 0.08$ and 0.04. The DE-RNN has 64 LSTM cells. Figure 1 shows the errors in the expectation and the standard deviation with respect to the analytical solution;

$$E_{y\sim p_T}[y_{t+1}|y_t] = y_t \exp(-0.5\delta t), \quad sd_{y\sim p_T}[y_{t+1}|y_t] = \sqrt{(0.5 + |y_t|)\delta_t}. \quad (22)$$

Here, $p_T$ denotes the true distribution of the CIR process. The normalized root mean-square errors (NRMSE) are defined as

$$e_\mu = \frac{\langle (E_{y\sim p_L}[y_{t+1}|y_t] - E_{y\sim p_T}[y_{t+1}|y_t])^2 \rangle^{1/2}}{\langle (y_t - E_{y\sim p_T}[y_{t+1}|y_t])^2 \rangle^{1/2}}, \quad (23)$$

$$e_\sigma = \frac{\langle (sd_{y\sim p_L}[y_{t+1}|y_t] - sd_{y\sim p_T}[y_{t+1}|y_t])^2 \rangle^{1/2}}{sd[y]}, \quad (24)$$

in which $\langle \cdot \rangle$ denotes an average over the testing data, $p_L$ is the distribution from the LSTM, and $sd[y]$ denotes the standard deviation of the data. The error in the expectation is normalized against a zeroth-order prediction, which assumes $y_{t+1} = y_t$.

In Figure 1, it is clearly shown that the prediction results are improved when a regularization is used to impose a smoothness condition. Comparing Figures 1 (a) and (b), for RCE, $e_\mu$ and $e_\sigma$ become smaller when a smaller $\delta y$ is used. As expected, $e_\sigma$ increases when $\lambda$ is large. But, for the smaller bin size, $\delta y = 0.04$, both $e_\mu$ and $e_\sigma$ are not so sensitive to $\lambda$. Similar to RCE, $e_\mu$ and $e_\sigma$ for CCE decrease at first as the penalty parameter $h$ increases. However, in general, RCE provides a better prediction compared to CCE.

NRMEs are listed in Table 1. For a comparison, the predictions by AR(1) and KF are shown. The CIR process is essentially a first-order autoregressive process. So, it is not surprising to see that AR(1) and KF, which are designed for the first-order AR process, outperforms DE-RNN for the prediction of the expectation. However, $e_\sigma$ of AR(1) and KF are much larger than that of DE-RNN, because those models assume an additive noise. The Gaussian process (GP) model has a relatively large $e_\mu$. But, GP outperforms AR(1) and KF in the prediction of the noise ($e_\sigma$). Still, $e_\sigma$ of RCE and CCE are less than 4%, while that of GP is about 10%, indicating that DE-RNN can model the complex noise process much better.

In Figure 2, a 200-step forecast by DE-RNN is compared with a Monte-Carlo solution of equation (21). DE-RNN is trained with $\delta y = 0.04$ and $\lambda = 200$. For the DE-RNN forecast, the testing data is supplied to DE-RNN for the first 100 time steps, i.e., for $t = -10$ to $t = 0$, and the SMC multiple-step forecast is performed for the next 200 time steps with 20,000 samples. It is shown that the multiple-step forecast by DE-RNN agrees very well with the MC solution of the CIR process. Note that, in Figure 2 (b), the noise process, as reflected in $sd[y_t]$, is a function of $y_t$, and hence the multi-step forecast of the noise increases rapidly first and then decreases before reaching a plateau.

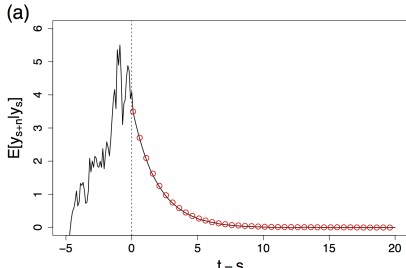 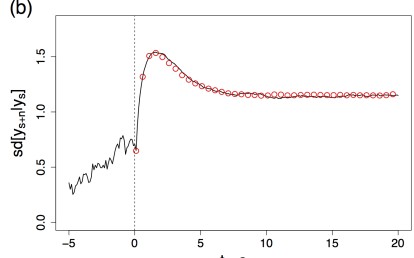

Figure 2: 200-step forecast of (a) expectation and (b) standard deviation of the CIR process. The circles denote the solution of Eqn (21) from a Monte Carlo method with $10^7$ samples.

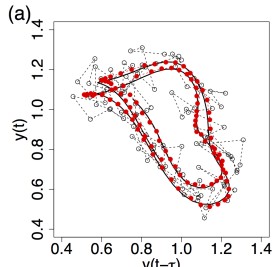 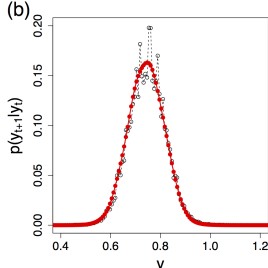

Figure 3: (a) Next-step prediction ($\bullet$) and the noisy observation ($\circ$) for the Mackey-Glass equation. The solid line denotes the ground truth, $y(t)$. (b) The next-step probability distribution, $p(\hat{y}_{t+1}|\hat{y}_t)$, from the standard CE ($\circ$) and CCE ($\bullet$) with $h = 5$.

The SMC forecast can accurately capture the behavior. Such kind of behavior can not be represented if a simple additive noise is assumed.

## 3.2 MACKEY-GLASS TIME SERIES

For the next test, we applied DE-RNN for a time series generated from the Mackey-Galss equation (Mackey & Glass, 1977);

$$\frac{dy}{dt} = \frac{\alpha y(t - \tau)}{1 + y^\beta (t - \tau)} - \gamma y(t). \tag{25}$$

We use the parameters adopted from Gers (2001), $\alpha = 0.2$, $\beta = 10$, $\gamma = 0.1$, and $\tau = 17$.

The Mackey-Glass equation is solved by using a third-order Adams-Bashforth method with a time step size of 0.02. The time series is generated by down-sampling, such that the time interval between consecutive data is $\delta t = 1$. A noisy observation is made by adding a white noise;

$$\hat{y}_t = y_t + \epsilon_t.$$

$\epsilon_t$ is a zero-mean Gaussian random variable with the noise level $sd[\epsilon_t] = 0.3 sd[y]$. A time series of the length $1.6 \times 10^5 \delta t$ is generated for the model trainig and another time series of length $2 \times 10^3 \delta t$ is made for the validation. DE-RNN is trained for $\delta y = 0.04 sd[y]$ and consists of 128 LSTM cells.

Figure 3 (a) shows the noisy observation and the expectation of the next-step prediction, $E[\hat{y}_{t+1}|\hat{y}_t]$, in a phase space. It is shown that DE-RNN can filter out the noise and reconstruct the original dynamics accurately. Even though the noisy data are used as an input, $E[\hat{y}_{t+1}|\hat{y}_t]$ accurately represents the original attractor of the chaotic system, indicating a strong de-noising capability of DE-RNN.

The estimated probability distribution is shown in Figure 3 (b). Without a regularization, the standard CE results in a noisy distribution, while the distribution from CCE shows a smooth Gaussian shape.

Table 2: NRMSEs of the Mackey-Galss time series. DE-RNN results are compared with autoregressive integrated moving average (ARIMA), Kalman filter (KF), and Gaussian process (GP) models.

| | RCE | | | | CCE | | ARIMA | KF | GP |
|---|---|---|---|---|---|---|---|---|---|
| | $\lambda = 0$ | $\lambda = 50$ | $\lambda = 100$ | $\lambda = 200$ | $h = 5$ | $h = 10$ | | | |
| $e_\mu$ | 0.198 | 0.196 | 0.198 | 0.196 | 0.185 | 0.198 | 0.662 | 0.903 | 0.327 |
| $e_\sigma$ | 0.032 | 0.023 | 0.027 | 0.038 | 0.013 | 0.020 | 0.191 | 0.351 | 0.048 |

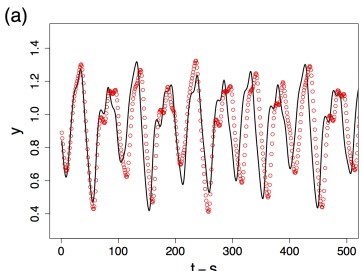
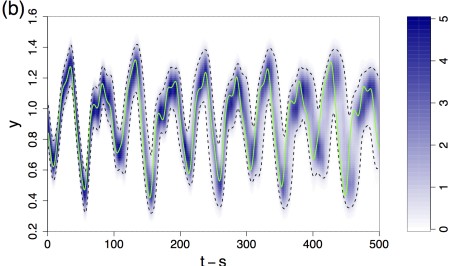

Figure 4: (a) 500-step forecast by a regression LSTM (○) and the ground truth (—). (b) The color contours denote a 500-step forecast of the probability distribution, $p(\hat{y}_{n+s}|\hat{y}_s)$, and the dashed lines are 95%-CI. The ground truth is shown as the solid line (—).

The prediction errors are shown in table 2. NRMSEs are defined as,

$$e_\mu = \frac{\langle (E[\hat{y}_{t+1}|\hat{y}_t] - y_{t+1})^2 \rangle^{1/2}}{\langle (\hat{y}_t - y_{t+1})^2 \rangle^{1/2}}, \quad e_\sigma = \frac{sd[\hat{y}_{t+1}|\hat{y}_t]}{sd[\epsilon_t]} - 1, \quad (26)$$

NRMSEs are computed with respect to the ground truth. Again, $e_\mu$ compares the prediction error to the zeroth-order prediction. In this example, the errors are not so sensitive to the regularization parameters. The best result is achieved by CCE. DE-RNN can make a very good estimation of the noise. The error in the noise component, $e_\sigma$, is only $2\% \sim 5\%$. Unlike the CIR process, NRMSEs from KF and ARIMA are much larger than those of DE-RNN. Because the underlying process is a delay-time nonlinear dynamical system, those linear models can not accurately approximate the complex dynamics. Since GP is capable of representing a nonlinear behavior of data, GP outperforms KF and ARIMA both in $e_\mu$ and $e_\sigma$. In particular, $e_\sigma$ of GP is similar to that of DE-RNN. However, $e_\mu$ of GP is about 1.5 times larger than DE-RNN.

A multiple-step forecast of the Mackey-Glass time series is shown in Figure 4. In the validation time series, the observations in $t \in [1, 100]\delta t$ are supplied to the DE-RNN to develop the internal state, and a 500-step forecast is made for $t \in [101, 600]\delta t$. In Figure 4 (a), it is shown that a multiple-step forecast by a standard regression LSTM approximates $y(t)$ very well initially, e.g, for $t < 80\delta t$, but eventually diverges for larger $t$. Because of the Mackey-Glass time series is chaotic, theoretically it is impossible to make a long time forecast. But, in the DE-RNN forecast, $y(t)$ is bounded by the 95%-confidence interval (CI) even for the 500-step forecast. Note that the uncertainty, denoted by 95%-CI grows only at a very mild rate in time. In fact, it is observed that CI is not a monotonic function of time. In DE-RNN, the 95%-CI may grow or decrease following the dynamics of the system, while for the conventional time series models, such as ARIMA and KF, the uncertainty is a non-decreasing function of time.

### 3.3 MAUNA LOA CO$_2$ OBSERVATION

In this experiments, DE-RNN is tested against the atmospheric CO$_2$ observation at Mauna Loa Observatory, Hawaii (Keeling et al., 2001). The CO$_2$ data set consists of weekly-average atmospheric CO$_2$ concentration from Mar-29-1958 to Sep-23-2017 (Figure 5 a). The data from Mar-29-1958 to Apr-01-2000 is used to train DE-RNN and a 17-year forecast is made from Apr-01-2000 to Sep-23-2017. This CO$_2$ data has been used in previous studies (Gal & Ghahramani, 2016; Rasmussen & Williams, 2006). In DE-RNN, 64 LSTM cells and $\delta y = 0.1sd[dy_t]$, in which $dy_t = y_{t+1} - y_t$, are used.

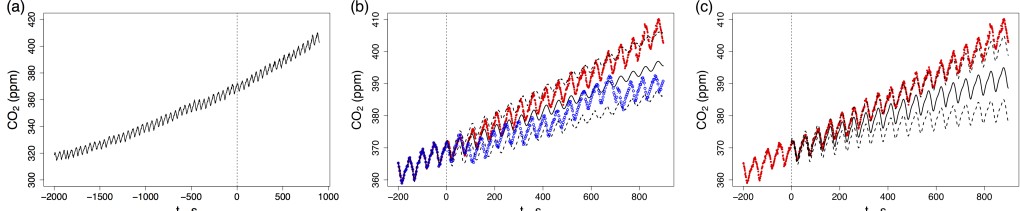

Figure 5: (a) Mauna Loa $CO_2$ observation. The vertical line denotes the boundary of the training and testing data. (b) 17-year forecast of the $CO_2$ concentration by DE-RNN: Apr-01-2000 Sep-23-2017. (c) 17-year forecast by Gaussian processes. The solid and dashed lines denote the expectation and 95%-CI, respectively. The observation is shown as the solid circles (●) and a regression LSTM is shown as the hollow circles (○) in (b). The time unit is a week.

Table 3: $l_\infty$ error for RCE, CCE and regression LSTM in °C.

| | RCE | CCE | LSTM |
|---|---|---|---|
| $\|E[\hat{y}_{n+s}\|\hat{y}_s] - \hat{y}_{n+s}\|_\infty$ | 4.61 | 2.70 | 9.48 |

The 17-year DE-RNN forecast, with 1,000 MC samples, is shown in Figure 5 (b). DE-RNN well represents the growing trend and the oscillatory patten of the $CO_2$ data. The $CO_2$ data is non-stationary, where the rate of increase of $CO_2$ is an increasing function of time. Since DE-RNN is trained against the history data, where the rate of $CO_2$ increase is smaller than the current, it is expected that the forecast will underestimate the future $CO_2$. $E[\hat{y}_{n+s}|\hat{y}_s]$ agrees very well with the observation for the first 200 weeks, but eventually underestimates $CO_2$ concentration. It is interesting to observe that the upper bound of the 95%-CI grows more rapidly than the expectation and provides a good approximation of the observation. For a comparison, the forecast by a regression LSTM is also shown. Similar to the chaotic Mackey-Glass time series, the regression LSTM makes a good prediction for a short time, e.g., $t < 100$ weeks, but eventually diverges from the observation. Note that the lower bound of 95%-CI encompasses the regression LSTM. Figure 5 (c) shows a forecast made by GP, following setup suggested by Rasmussen & Williams (2005). For a short-term forecast ($< 50$ weeks), GP provides a sharper estimate of the uncertainty, i.e., a smaller 95%-CI interval. However, even for a mid-term forecast, $100 \sim 600$ weeks, the ground truth is close or slightly above the upper bound of 95%-CI. Because of the different behaviors, it is difficult to conclude which method, DE-RNN or GP, provides a better forecast. But, it should be noted that the GP is based on a combination of handcrafted kernels specifically designed for this particular problem, while such careful tuning is not required for DE-RNN.

### 3.4 CPU TEMPERATURE FORECAST

In the last experiment, IBM Power System S822LC and NAS Parallel Benchmark (NPB) are used to generate the temperature trace. Figure 6 (a) shows the temperature of a CPU. The temperature sensor generates a discrete data, which has a resolution of 1°C. The CPU temperature is controlled by three major parameters; CPU frequency, CPU utilization, and cooling fan speed. In this experiment, we have randomized the CPU frequencies and job arrival time to mimic the real workload behavior, while the fan speed is fixed to 3300RPM. The time step size is $\delta t = 2$ seconds. Accurate forecast of CPU temperature for a future workload scenario is essential in developing an energy-efficient control strategy for the thermal management of a cloud system.

Figure 6 (c) and (d) show multiple-step forecasts of the CPU temperature by RCE and a regression LSTM, respectively. The bin size is $\delta y = 0.18$°C, which is smaller than the sensor resolution. In the forecast, the future control parameters are given to DE-RNN. In other words, DE-RNN predicts the probability distribution of future temperature with respect to a control scenario, i.e., $p(\hat{y}_{t+n}|\widehat{Y}_{0:t}, U_{0:t+n-1})$. The forecast is made by using 5,000 Monte Carlo samples. Here, 1800-step forecast is made, $t = 0 \sim 3,600$ sec. and only the results in $t \in (50, 1800)$ sec. is shown. While the regression LSTM makes a very noisy prediction near the local peak temperature at $t \simeq 500$, RCE provides a much more stable forecast. Table 3 shows the $l_\infty$-errors, i.e., maximum absolute

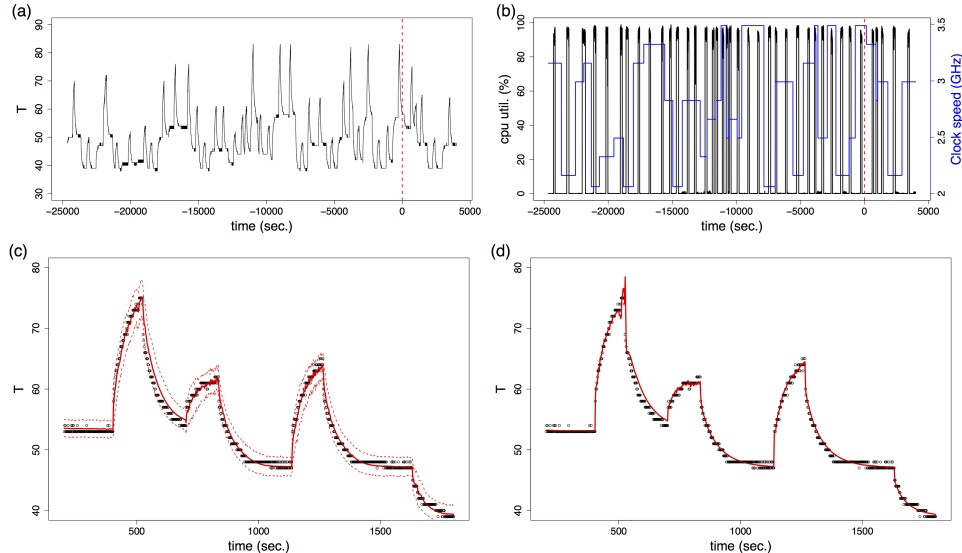

Figure 6: (a) Temperature of a CPU in °C. (b) Control parameters; CPU utilization (black) and Clock speed (blue). Multiple-step forecasts by (c) RCE and (d) regression LSTM. The solid and dashed lines in (c) denote $E[\hat{y}_{n+s}|\hat{y}_s]$ and 95%-CI, respectively. The circles are the observations.

difference. The maximum error is observed near the peak temperature at $t \simeq 500$. ARIMA, KF, and GP are also tested for the multiple-step forecast, but the results are not shown because their performance is much worse than the LSTMs. The changes in the temperature are associated with step changes of some control parameters. Such abrupt transitions seem to cause the large oscillation in the regression LSTM prediction. But, for RCE or CCE, the prediction is made from an ensemble of Monte Carlo samples, which makes it more robust to such abrupt changes. Also, note that for $t < 200$ sec., RCE prediction ($\simeq 53.4$°C) is in between the two discrete integer levels, $53$°C and $54$°C, which correctly reflects the uncertainty in the measurement, while the regression LSTM ($\simeq 53.1$°C) more closely follows one of the two levels.

### 3.5 LORENZ TIME SERIES

Finally, in this experiment we evaluate the performance of the multivariate DE-RNN, for which we used a noisy multivariate time series generated by the Lorenz equations (Lorenz, 1963)

$$\frac{dy^{(1)}}{dt} = \alpha_1(y^{(2)} - y^{(1)}),$$
$$\frac{dy^{(2)}}{dt} = y^{(1)}(\alpha_2 - y^{(3)}) - y^{(2)}, \qquad (27)$$
$$\frac{dy^{(3)}}{dt} = y^{(1)}y^{(2)} - \alpha_3 y^{(3)}.$$

We used the coefficients from Lorenz (1963), which are $\alpha_1 = 10$, $\alpha_2 = 8/3$, and $\alpha_3 = 28$. The system of equations (27) is solved by using a third-order Adams-Bashforth method with a time step size of 0.001 and a time series data set is generated by downsampling, such that $\delta t = 0.02$. A multivariate joint normal distribution is added to the ground truth to generate a noisy time series, i.e.,

$$\begin{pmatrix} \hat{y}_t^{(1)} \\ \hat{y}_t^{(2)} \\ \hat{y}_t^{(3)} \end{pmatrix} = \begin{pmatrix} y_t^{(1)} \\ y_t^{(2)} \\ y_t^{(3)} \end{pmatrix} + \boldsymbol{\epsilon}_t, \quad \boldsymbol{\epsilon}_t = \mathcal{N}(\mathbf{0}, \boldsymbol{\Sigma}), \text{ and } \boldsymbol{\Sigma} = \begin{pmatrix} \sigma_1^2 & 0.25\sigma_1\sigma_2 & -0.25\sigma_1\sigma_3 \\ & \sigma_2^2 & -0.25\sigma_2\sigma_3 \\ & & \sigma_3^2 \end{pmatrix}. \qquad (28)$$

Here, the noise level is set to $\sigma_1 = 0.2sd[y^{(1)}]$, $\sigma_2 = 0.3sd[y^{(2)}]$, and $\sigma_3 = 0.2sd[y^{(3)}]$. Three DE-RNNs are trained to model $p(\hat{y}_{t+1}^{(1)}|\hat{\boldsymbol{Y}}_{0:t})$, $p(\hat{y}_{t+1}^{(2)}|\hat{y}_{t+1}^{(1)}, \hat{\boldsymbol{Y}}_{0:t})$, and $p(\hat{y}_{t+1}^{(3)}|\hat{y}_{t+1}^{(2)}, \hat{y}_{t+1}^{(1)}, \hat{\boldsymbol{Y}}_{0:t})$, re-

Table 4: Normalized errors of the Lorenz time series. DE-RNN results are compared with vector autoregressive (VAR) and Gaussian process (GP) models.

|  | $e_\mu$ | $e_\sigma$ | $e_{\Sigma_{12}}$ | $e_{\Sigma_{13}}$ | $e_{\Sigma_{23}}$ |
|---|---|---|---|---|---|
| CE | 0.19 | 0.027 | 0.146 | -0.012 | 0.024 |
| RCE | 0.20 | 0.030 | 0.155 | -0.010 | 0.026 |
| CCE | 0.20 | 0.018 | 0.123 | -0.009 | 0.022 |
| VAR | 1.00 | 0.318 | 1.485 | 0.483 | 0.649 |
| GP | 0.58 | 0.296 | - | - | - |

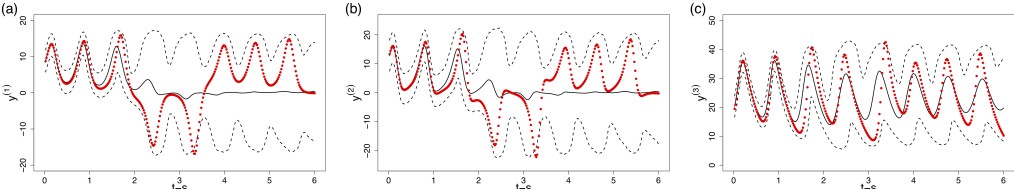

Figure 7: Multiple-step forecast of the Lorenz time series by DE-RNN trained with CCE. The solid and dashed lines denote the expectation and 95%-CI, respectively. The ground truth is shown as sold circles.

spectively, with the grid spaces $\delta y^{(i)} = 0.02 sd[\hat{y}^{(i)}]$ for $i = 1, 2, 3$. Each DE-RNN has 128 LSTM cells. The penalty parameters for RCE and CCE are $\lambda = 50$ and $h = 5$, respectively.

Figure 4 shows normalized errors for the next-step prediction. The normalized errors are defined as

$$e_\mu = \sqrt{\frac{1}{3}\sum_{i=1}^{3}\frac{\langle(E[\hat{y}_{t+1}^{(i)}] - y_{t+1}^{(i)})^2\rangle}{\langle(\hat{y}_t^{(i)} - y_{t+1}^{(i)})^2\rangle}}, \ e_\sigma = \sqrt{\frac{1}{3}\sum_{i=1}^{3}\frac{Var[\hat{y}_{t+1}^{(i)}]}{\sigma_i^2}} - 1, \ e_{\Sigma_{ij}} = \frac{Cov[\hat{y}_{t+1}^{(i)}, \hat{y}_{t+1}^{(j)}]}{\Sigma_{ij}} - 1.$$

(29)

The moments of the joint PDF are computed by the Monte Carlo method with a sample size of $5 \times 10^3$. It is shown that DE-RNN makes a very good prediction of both expectations and covariances. The error in the covariance is less than 4% except for $\Sigma_{12}$. It is shown that DE-RNN is able to make correct predictions of not only the magnitude, but also the signs of the covariance. A vector autoregressive model (VAR) also predicts the signs of the covariance correctly. But, the errors in the expectation and covariances are much larger than DE-RNN. The GP used in the experiment assumes an independent noise, i.e. $\boldsymbol{\Sigma} = \rho^2 \boldsymbol{I}$. Hence, $e_{\Sigma_{ij}}$ is not evaluated for GP. Similar to the Mackey-Glass time series, GP outperforms VAR, but the errors are larger than DE-RNN. The error in $e_\sigma$ is about 10 times larger than DE-RNN, while $e_\mu$ of GP is about 3 times larger than DE-RNN.

Figure 7 shows 300-step forecasts of the Lorenz time series. The expectations from DE-RNNs make a good prediction of the ground truth up to $t - s < 1.5$. Then, the expectations start to diverge from the ground truth, which is accompanied by a sudden increase in the 95%-CI. For a longer-time forecast, e.g., $t - s > 4$, the 95%-CI exhibits an oscillatory patten depending on the dynamics of the Lorenz system and well captures the oscillations in the ground truth.

## 4   CONCLUDING REMARKS

We present DE-RNN to compute the time evolution of a probability distribution for complex time series data. DE-RNN employs LSTM to learn multiscale, nonlinear dynamics from the noisy observations, which is supplemented by a softmax layer to approximate a probability density function. To assign probability to the softmax output, we use a mapping from $\mathbb{R}$ to $\mathbb{N}_+$, which leads to a cross-entropy minimization problem. To impose a geometric structure in the distribution, two regularization strategies are proposed. The regularized cross-entropy method is analogous to the penalized maximum likelihood estimate for the density estimation, while the convolution cross-entropy method is motivated by the kernel density estimation. The proposed algorithm is validated against

three synthetic data set, for which we can compare with the analytical solutions, and two real data sets.

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

## APPENDIX A    MULTIVARIATE DE-RNN

Recall the product rule from Section 2.3

$$p(\hat{\boldsymbol{y}}_{t+1}) = p\left(\hat{y}_{t+1}^{(l)}|\hat{y}_{t+1}^{(l-1)}, \cdots, \hat{y}_{t+1}^{(1)}\right) p\left(\hat{y}_{t+1}^{(l-1)}|\hat{y}_{t+1}^{(l-2)}, \cdots, \hat{y}_{t+1}^{(1)}\right) \cdots p\left(\hat{y}_{t+1}^{(2)}|\hat{y}_{t+1}^{(1)}\right) p\left(\hat{y}_{t+1}^{(1)}\right).$$

The extension of DE-RNN algorithm for univariate data to an $l$-dimensional multivariate time series is straightforward. First, define the discretization grid points, $\boldsymbol{\alpha}^{(i)} = (\alpha_0^{(i)}, \cdots, \alpha_{K_i}^{(i)})$, for every variable, i.e., $i = 1, \cdots, l$. Here, $K_i$ is the number of the discretization intervals for the $i$-th variable. Then, we can define the discretization intervals, $\mathcal{I}_j^{(i)} = (\alpha_{j-1}^{(i)}, \alpha_j^{(i)})$, and the mapping functions, $\mathcal{C}^{(i)}(y^{(i)})$, for the $l$ variables.

The first component of the product rule is the marginal PDF, $p(\hat{y}_{t+1}^{(1)})$. Hereafter, the obvious dependency on $\widehat{\boldsymbol{Y}}_{0:t}, \boldsymbol{U}_{0:t}$ in the notation is omitted for simplicity. The marginal PDF can be computed by the same method as for the univariate time series. To train DE-RNNs for the conditional PDFs for the $i$-th variable, $p(\hat{y}_{t+1}^{(i)}|\hat{y}_{t+1}^{(i-1)}, \cdots \hat{y}_{t+1}^{(1)})$, the original time series data of length $N$, $\boldsymbol{D}_R = \{\hat{\boldsymbol{y}}_t; \hat{\boldsymbol{y}}_t \in \mathbb{R}^l, \text{and } t = 1, \ldots, N\}$, is discretized by using $\mathcal{C}^{(i)}$, which gives us $\boldsymbol{D}_C^{(i)} = \{(c_t^{(i)}, \hat{\boldsymbol{y}}_t); c_t^{(i)} \in \mathbb{N}_+, \hat{\boldsymbol{y}}_t \in \mathbb{R}^l, \text{and } t = 1, \ldots, N\}$, where $c_t^{(i)} = \mathcal{C}^{(i)}(\hat{y}_t^{(i)})$. Then the output softmax layer, $\boldsymbol{P}_{t+1}^{(i)}$, is computed by an LSTM as

$$\boldsymbol{P}_{t+1}^{(i)} = \Psi_d \circ \Psi_e(\hat{y}_{t+1}^{(i-1)}, \cdots, \hat{y}_{t+1}^{(1)}, \hat{\boldsymbol{y}}_t, \boldsymbol{s}_t^{(i)}), \tag{30}$$

in which $\boldsymbol{s}_t^{(i)}$ is the internal state of the $i$-th variable. In other words, in the training of the DE-RNN for $p(\hat{y}_{t+1}^{(i)}|\hat{y}_{t+1}^{(i-1)}, \cdots \hat{y}_{t+1}^{(1)})$, the input vector is the variables at the current time step, $\hat{\boldsymbol{y}}_t$, combined with the conditioning variables in the next step, $(\hat{y}_{t+1}^{(1)}, \cdots, \hat{y}_{t+1}^{(i-1)})$, and the target is the class label, $c_{t+1}^{(i)}$, in the next time step. The DE-RNN can be trained by minimizing RCE or CCE as described in Section 2.2. Observe that during the training phase, each DE-RNN is independent from each other. Therefore, all the DE-RNNs can be trained in parallel, significantly improving the computational efficiency, enabling the method to scale only linearly with the number of dimensions $l$.

Once all the conditional PDFs are obtained, the joint PDF can be computed by a product of the DE-RNN outputs. For a demonstration, the covariance of a bivariate time series can be computed as

$$Cov\left(\hat{y}_{t+1}^{(1)}, \hat{y}_{t+1}^{(2)}\right) = \iint \hat{y}_{t+1}^{(1)} \hat{y}_{t+1}^{(2)} p\left(\hat{y}_{t+1}^{(2)}|\hat{y}_{t+1}^{(1)}, \widehat{\boldsymbol{Y}}_{0:t}\right) p\left(\hat{y}_{t+1}^{(1)}|\widehat{\boldsymbol{Y}}_{0:t}\right) d\hat{y}_{t+1}^{(2)} d\hat{y}_{t+1}^{(1)}$$

$$\simeq \sum_{i=1}^{K_1} \left\{ \alpha_{i-1/2}^{(1)} P_i^{(1)}(\hat{\boldsymbol{y}}_t) \sum_{j=1}^{K_2} \alpha_{j-1/2}^{(2)} P_j^{(2)}\left(\alpha_{i-1/2}^{(1)}, \hat{\boldsymbol{y}}_t\right) \right\}, \tag{31}$$

where the time index, $(t+1)$, is omitted in the notation of the softmax output, $\boldsymbol{P}_j^{(i)}$, and the subscript $j$ denotes the $j$-th element of the softmax layer. Note that, although there is no dependency between DE-RNNs during training, in the prediction phase of computing the joint PDF, there is a hierarchical dependency between all the DE-RNNs. This kind of direct numerical integration does not scale well for number of dimensions $l \gg 1$. For a high dimensional system, a sparse grid (Xiu & Hesthaven, 2005) or a Monte Carlo method can be used to speed up the numerical integration. We outline a Monte Carlo procedure in Algorithm 2. Comparing with Algorithm 1, an extension of Algorithm 2 for a multiple-step forecast of a multivariate time series is straightforward.

## APPENDIX B    DE-RNN ACHITECTURE

In this Section we present a few extra details of the proposed DE-RNN algorithm. Figure 8 shows the architecture of DE-RNN as was used during the experiments in Section 3. In Figure 9 we show the process of computing one-step-ahead predictions of univariate time series as was presented in Section 2.2. Note that since DE-RNN estimates approximation of the predictive probability distribution, the predicted value, e.g., for time step $t+1$, is the discrete approximation of $E[\hat{y}_{t+1}|\widehat{\boldsymbol{Y}}_{0:t}, \boldsymbol{U}_{0:t}]$, i.e., the expectation of $\hat{y}_{t+1}$ given all the observations and control inputs up to time $t$. Finally, in

---

**Algorithm 2** Monte Carlo method for the computation of the joint PDF

---

**Input**: $\widehat{\boldsymbol{Y}}_{0:t}$, and number of Monte Carlo samples, $N_s$

**Output**: $p(\hat{\boldsymbol{y}}_{t+1}|\widehat{\boldsymbol{Y}}_{0:t})$ (density estimation from $\hat{\boldsymbol{y}}_{t+1}$)

Initialization: Set LSTM states to $\boldsymbol{s}_0 = \boldsymbol{h}_0 = \boldsymbol{0}$

Perform a sequential update of LSTM up to time $t-1$ from the noisy observations ($\widehat{\boldsymbol{Y}}_{0:t}$):

**for** $i = 1, t-1$ **do**

$$\boldsymbol{s}_i^{(1)} = \Psi_e(\hat{\boldsymbol{y}}_i, \boldsymbol{s}_{i-1}^{(1)})$$

  **for** $j = 2, l$ **do**

$$\boldsymbol{s}_i^{(j)} = \Psi_e(\hat{y}_{i+1}^{(1)}, \cdots, \hat{y}_{i+1}^{(j-1)}, \hat{\boldsymbol{y}}_i, \boldsymbol{s}_{i-1}^{(j)})$$

  **end for**
**end for**

Monte Carlo sampling for $p(\hat{\boldsymbol{y}}_{t+1}|\widehat{\boldsymbol{Y}}_{0:t})$:

Compute the predictive distribution of $\hat{y}_{t+1}^{(1)}$

$$\boldsymbol{P}_{t+1}^{(1)} = \Psi_d \circ \Psi_e(\hat{\boldsymbol{y}}_t, \boldsymbol{s}_{t-1}^{(1)})$$

**for** $n = 1, N_s$ **do**

  Draw a sample, $\hat{y}_{t+1}^{(1),n}$, from $\boldsymbol{P}_{t+1}^{(1)}$.

  **for** $j = 2, l$ **do**

    Compute the conditional distribution for $\hat{y}_{t+1}^{(j)}$ for each sample

$$\boldsymbol{P}_{t+1}^{(j),n} = \Psi_d \circ \Psi_e(\hat{y}_{t+1}^{(1),n}, \cdots, \hat{y}_{t+1}^{(j-1),n}, \hat{\boldsymbol{y}}_t, \boldsymbol{s}_{t-1}^{(j)})$$

    Draw a sample, $\hat{y}_{t+1}^{(j),n}$, from $\boldsymbol{P}_{t+1}^{(j),n}$

  **end for**
**end for**

---

Figure 10 we show the details of the multi-step forecast for univariate time series as was presented in Algorithm 1, in Section 2.4. Using sequential Monte Carlo method, the discrete approximation of the predictive distribution $p(\hat{y}_{t+n}|\widehat{\boldsymbol{Y}}_{0:t}, \boldsymbol{U}_{0:t+n-1})$ is estimated using $N_s$ samples.

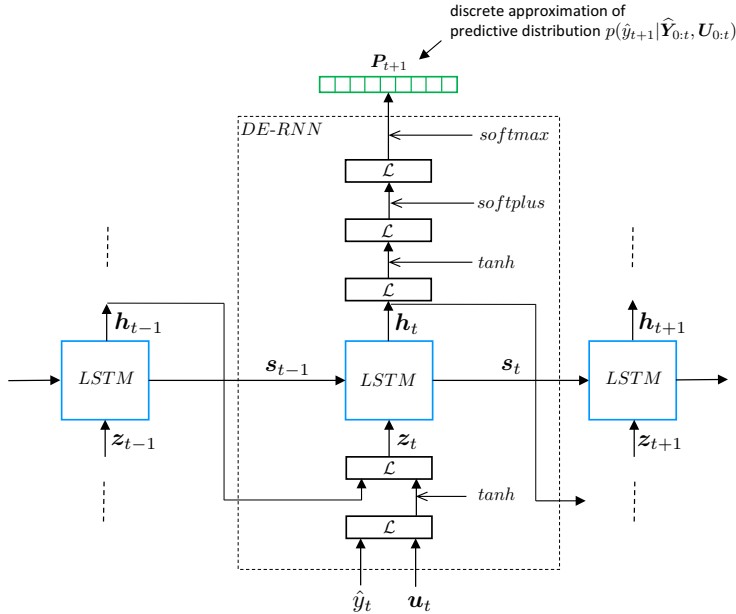

Figure 8: Architecture of the DE-RNN algorithm.

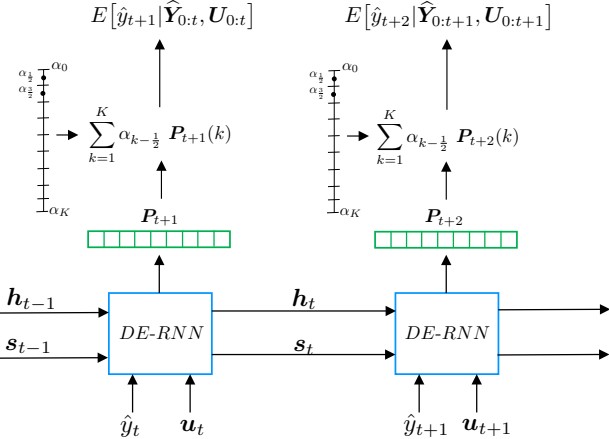

Figure 9: Details of the computation for the one-step-ahead predictions. At a given step the model computes a discrete approximation of the expectation for the next step observation.

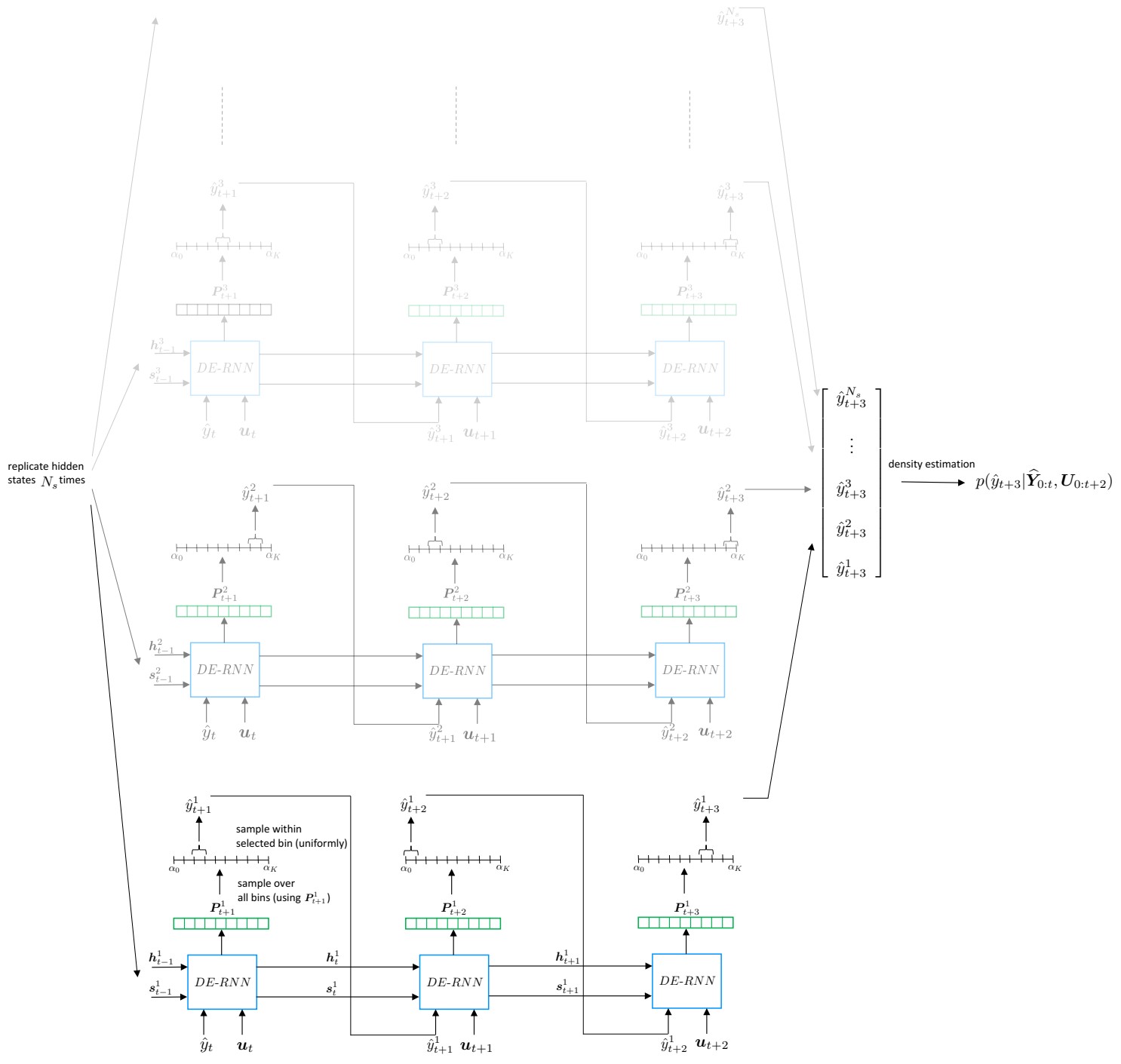

Figure 10: Visualization of the multi-step-ahead forecast made by DE-RNN. The shown illustration is for computing 3-step-ahead predictive probability distribution $p(\hat{y}_{t+3}|\widehat{\boldsymbol{Y}}_{0:t}, \boldsymbol{U}_{0:t+2})$. $Ns$ independent DE-RNN instances, $i = 1, \ldots, N_s$, are initialized identically with the hidden states from the time step of the last observation. Each DE-RNN instance $i$ uses a two-stage sampling procedure to propagate its own value of $\hat{y}^i$ in time. At the end, $N_s$ samples of $\hat{y}^i_{t+3}$ are used to compute a discrete approximation of the predictive distribution $p(\hat{y}_{t+3}|\widehat{\boldsymbol{Y}}_{0:t}, \boldsymbol{U}_{0:t+2})$.

