# OpenReview forum: "Learning temporal evolution of probability distribution with Recurrent Neural Network"
_ICLR.cc/2018/Conference — Reject_

### Official Review · AnonReviewer3 · 2017-11-19
**Interesting ideas that extend LSTM to produce probabilistic forecasts for univariate time series, experiments are okay. Unclear if this would work at all in higher-dimensional time series. It is also unclear to me what are the sources of the uncertainties captured.**

**Rating:** 6
**Confidence:** 2

**Review:**

Interesting ideas that extend LSTM to produce probabilistic forecasts for univariate time series, experiments are okay. Unclear if this would work at all in higher-dimensional time series. It is also unclear to me what are the sources of the uncertainties captured.


The author proposed to incorporate 2 different discretisation techniques into LSTM, in order to produce probabilistic forecasts of univariate time series. The proposed approach deviates from the Bayesian framework where there are well-defined priors on the model, and the parameter uncertainties are subsequently updated to incorporate information from the observed data, and propagated to the forecasts. Instead, the conditional density p(y_t|y_{1:t-1|, \theta}) was discretised by 1 of the 2 proposed schemes and parameterised by a LSTM. The LSTM was trained using discretised data and cross-entropy loss with regularisations to account for ordering of the discretised labels. Therefore, the uncertainties produced by the model appear to be a black-box. It is probably unlikely that the discretisation method can be generalised to high-dimensional setting?

Quality: The experiments with synthetic data sufficiently showed that the model can produce good forecasts and predictive standard deviations that agree with the ground truth. In the experiments with real data, it's unclear how good the uncertainties produced by the model are. It may be useful to compare to the uncertainty produced by a GP with suitable kernels. In Fig 6c, the 95pct CI looks more or less constant over time. Is there an explanation for that?

Clarity: The paper is well-written. The presentations of the ideas are pretty clear.

Originality: Above average. I think the regularisation techniques proposed to preserve the ordering of the discretised class label are quite clever.

Significance: Average. It would be excellent if the authors can extend this to higher dimensional time series.

I'm unsure about the correctness of Algorithm 1 as I don't have knowledge in SMC.

---

> ### Author Response · Authors · 2017-12-28
> **Responses and Revisions**
>
> We thank the referee for carefully reading our manuscript and providing helpful comments.
>
> 1. Estimated uncertainty: We are aware of the previous studies, notably Kendall & Gal (2017) and Gal & Ghahramani (2015), where both the model (epistemic) and data (aleatoric) uncertainties are carefully studied in a Bayesian framework. However, as pointed out by the referee, we approach the problem from the Frequentist framework. We aim to make an inference of the probability distribution of the data, or aleatoric uncertainty, given the accuracy of the model. The estimated predictive probability distribution will model both: the data probability distribution and the model error.  However, once the model is powerful enough and the data size is large enough, the estimated probability distribution converges to the true distribution of the data, meaning that the estimated uncertainties will represent the noise in the data. In fact, this latter situation is shown to be indeed the case during the evaluations on synthetic data when powerful enough RNN model is employed.
>
> 2. Multivariate time series. We agree with the referee that a naïve extension of the DE-RNN to a higher-dimension based on the tensor-product space will not be scalable. Instead, we propose to compute the joint probability distribution by using a product rule. The detailed method is presented in Section 2.3 and Appendix A. A new numerical experiment is shown in Section 3.5. The proposed DE-RNN for a multivariate time series scales linearly with the number of the dimensions.
>
> 3. Gaussian Processes: We added new comparisons with Gaussian process models. We also used a GP model for the multiple-step forecast on CO2 and CPU temperature data sets. However we did not include the results of GP on the CPU problem since it did not perform well, which might be due the presence of discrete input variables (CPU utilization and clock speed) for which GP is not a suitable approach.
>
> 4. Question regarding the uncertainty bound in Fig. 6c: In a multiple-step forecast, the time evolution of a probability density function is essentially a diffusion process. So, in general, it is expected that the prediction uncertainty, represented by 95%-CI, increase with the forecast horizon, which is the case for most of the conventional time series prediction models. But, if we look at the so called “master equation” of the time evolution of probability density function (PDF), or the Fokker-Planck equation, the time evolution of a PDF is determined by two terms, advection in the probability space and the regular diffusion (Brownian) process. The later makes the uncertainty, or the width of a PDF, grows in time. However, it seems that when an RNN is used to model the time series, it makes the first term (advection term) convergent, which counteracts the diffusion process. We have observed (see, for example, figure 3 a), when a noisy input data is given to an RNN, surprisingly the prediction seems always moving toward the ground truth. We have shown that the prediction error with respect to the ground truth becomes smaller than the noise level. This observation suggests that the convergence to the ground truth counterbalances the diffusion by the random process, which explains why the uncertainty bound is no longer a monotonically increasing function of a forecast horizon. Although it is not shown, we have tested DE-RNN for a forced Van der Pol oscillator, which has a stationary state, similar to the experiment in section 3.4. For this kind of deterministic system, it was found that the uncertainty bounds fluctuate but do not grow even for a very long (2,000-step) forecast.

---

### Official Review · AnonReviewer1 · 2017-11-27

**Rating:** 5
**Confidence:** 4

**Review:**

The papers proposes a recurrent neural network-based model to learn the temporal evolution of a probability density function. A Monte Carlo method is suggested for approximating the high dimensional integration required for multi-step-ahead prediction.

The approach is tested on two artificially generated datasets and on two real-world datasets, and compared with standard approaches such as the autoregressive model, the Kalman filter, and a regression LSTM.

The paper is quite dense and quite difficult to follow, also due to the complex notation used by the authors.

The comparison with other methods is very week, the authors compare their approach with two very simple alternatives, namely a first-order autoregressive mode and the Kalman filter.  More sophisticated should have been employed.

---

> ### Author Response · Authors · 2017-12-28
> **Responses and Revisions**
>
> The major purpose of this study is to introduce a new framework to compute the probability distribution of a time series and to compute a time evolution of the probability distribution in the future, which has a direct relevance to many applications in the modeling of physical or industrial processes. Hence, we focused more on stochastic processes with underlying (physical) dynamics systems.
>
> 1. Synthetic and real data: Unfortunately, in this application area, most of the data are proprietary or confidential and there are only a limited number of publicly accessible data set for this kind of modeling. Therefore, we focused on synthetic data for thorough model validation and testing. Although we agree that the behavior of the synthetic data may not be exactly replicated in a real problem, the use of synthetic data allows us to have a deeper investigation into the behavior of the model (DE-RNN) under various conditions. Nevertheless, we also used two real data sets and these experiments similarly showed the advantage of our method over the traditional approaches.
>
> 2. As pointed out by the referee, we have added new comparison results by using Gaussian processes in sections 3.1 ~ 3.3 and also a new experiment for a multivariate time series in section 3.5.

---

### Official Review · AnonReviewer4 · 2017-12-06

**Rating:** 6
**Confidence:** 4

**Review:**

This work proposes an LSTM based model for time-evolving probability densities. The model does not assume an explicit prior over the underlying dynamical systems, instead only uncertainty over observation noise is explicitly considered. Experiments results are good for given synthetic scenarios but less convincing for real data.

Clarity: The paper is well-written. Some notations in the LSTM section could be better explained for readers who are unfamiliar with LSTMs. Otherwise, the paper is well-structured and easy to follow.

Originality: I'm not familiar with LSTMs, it is hard for me to judge the originality here.

Significance: Average. The work would be stronger if the authors can extend this to higher dimensional time series. There are also many papers on this topic using Gaussian process state-space (GP-SSM) models where an explicit prior is assumed over the underlying dynamical systems. The authors might want to comment on the relative merits between GP-SSMs and DE-RNNs.

The SMC algorithm used is a sequential-importance-sampling (SIS) method. I think it's correct but may not scale well with dimensions.

---

> ### Author Response · Authors · 2017-12-28
> **Responses and Revisions**
>
> We thank the referee for carefully reading our manuscript and providing helpful feedback.
>
> 1. GP-SSM: We thank the referee for bringing GP-SSM to our attention. As suggested by the referee, we added a comment about GP-SSM in the literature survey.
>
> 2. Multivariate time series: We agree with the referee that directly extending the current DE-RNN for a multivariate time series, based on a tensor-product approach, will not be scalable. Instead, we presented a new method to compute the joint probability distribution by using a product rule (section 2.3 and Appendix A). The new method relies on a product of independently trained DE-RNNs to compute the joint probability distribution. The computational complexity of this method increases linearly with the number of the dimension. A new numerical experiment for the multivariate time series is shown in section 3.5.

---

### Decision · Program_Chairs · 2018-01-29
**ICLR 2018 Conference Acceptance Decision**

**Decision:**

Reject

**Comment:**

Thank you for submitting you paper to ICLR. Two of the reviewers are concerned that the paper’s contributions are not significant enough —either in terms of the theoretical or experimental contribution -- to warrant publication. The authors have improved the experimental aspect to include a more comprehensive comparison, but this has not moved the reviewers.